# Simple Preparation of Polydimethylsiloxane and Polyurethane Blend Film for Marine Antibiofouling Application

**DOI:** 10.3390/polym13142242

**Published:** 2021-07-08

**Authors:** Jirasuta Chungprempree, Sutep Charoenpongpool, Jitima Preechawong, Nithi Atthi, Manit Nithitanakul

**Affiliations:** 1The Petroleum and Petrochemical College, Chulalongkorn University, Chula Soi 12, Wangmai Pathumwan, Bangkok 10330, Thailand; jirasuta.chung@gmail.com (J.C.); sutep.cha@gmail.com (S.C.); jitima.pre@gmail.com (J.P.); 2Center of Excellence on Petrochemical and Materials Technology, Chula Soi 12, Wangmai Pathumwan, Bangkok 10330, Thailand; 3Thai Microelectronics Center (TMEC), National Electronics and Computer Technology Center (NECTEC), Chachoengsao 24000, Thailand; nithi.atthi@nectec.or.th

**Keywords:** antifouling, polymer blend, polydimethylsiloxane elastomer, polyurethane, hydrophobic film

## Abstract

A key way to prevent undesirable fouling of any structure in the marine environment, without harming any microorganisms, is to use a polymer film with high hydrophobicity. The polymer film, which was simply prepared from a blend of hydrophobic polydimethylsiloxane elastomer and hydrophilic polyurethane, showed improved properties and economic viability for antifouling film for the marine industry. The field emission scanning electron microscope and energy dispersive X-ray spectrometer (FESEM and EDX) results from the polymer blend suggested a homogenous morphology and good distribution of the polyurethane disperse phase. The PDMS:PU blend (95:5) film gave a water contact angle of 103.4° ± 3.8° and the PDMS film gave a water contact angle of 109.5° ± 4.2°. Moreover, the PDMS:PU blend (95:5) film could also be modified with surface patterning by using soft lithography process to further increase the hydrophobicity. It was found that PDMS:PU blend (95:5) film with micro patterning from soft lithography process increased the contact angle to 128.8° ± 1.6°. The results from a field test in the Gulf of Thailand illustrated that the bonding strength between the barnacles and the PDMS:PU blend (95:5) film (0.07 MPa) were lower than the bonding strength between the barnacles and the carbon steel (1.16 MPa). The barnacles on the PDMS:PU blend (95:5) film were more easily removed from the surface. This indicated that the PDMS:PU blend (95:5) exhibited excellent antifouling properties and the results indicated that the PDMS:PU blend (95:5) film with micro patterning surface could be employed for antifouling application.

## 1. Introduction

The surfaces of buildings and boats in marine environments tend to accumulate high concentrations of fouling attachments of a micro–macro organism biofilm. This is the major cause of damages to structures and equipment in a marine environment [1]. In general, biofouling is formed due to the attachment of micro-organisms (bacteria, algae and fungi) and macro-organisms (barnacles, sponges, seaweed, etc.) onto the structures’ surfaces in water [2]. Especially on cargo ships, biofouling is a major problem because it increases the weight, reduces the ship speed, leads to a rise in fuel consumption of up to 40%, and increases overall costs by up to 77% [3] and also carries non-native species to the other marine environments. Moreover, medical equipment should also avoid the attachment of microorganisms, which is the cause of middle ear infections, kidney infections, dental caries, urinary tract infections and chronic prostatitis [4]. Biofouling is currently a major focus for the marine and medical industries. 

Tributyltin was one of the most widely used biocides for antifouling paints [5]. Many researchers have found negative effects of tributyltin due to its toxicity to other marine organisms [5] for example, it induces imposex and intersex conditions in marine snails via mechanisms of endocrine disruption. In 2003, the international maritime organization (IMO) banned tributyltin from any antifouling application [6]. Then, antifouling paint was developed using copper oxide pigment, thiocyanate, cuprous bromide, etc. Copper oxide was selected over the other pigments due to being inexpensive and soluble [7]; however, it is extremely toxic at elevated concentrations and negatively impacts mussels, fish, and crustaceans. Its mechanisms of toxicity include interference with osmoregulation due to enzyme inhibition, decreasing immune function and decreasing respiration [8]. In recent years, industry has focused on developing a non-toxic antifouling coating. There are several reports which studied the non-toxic antifouling properties of polymeric coatings such as polyethylene glycol (PEG) [9], fluoropolymers [9], and xerogels [10].

Polydimethylsiloxane (PDMS) is widely used for antifouling coating because it consists of -Si-O-Si- and a side chain of the -CH_3_ group, leading to good thermal stability, elasticity, and hydrophobicity [11]. The polymer main chain of siloxane bonds (-Si-O-Si-) has a high bond energy and bond angle that provide good thermal stability and elasticity for PDMS [12]. Side chain groups of PDMS are highly hydrophobic (water contact angle (WCA) 107–110°) [13] and non-polar groups, which give a low surface energy as well as being excellent for releasing fouling materials [14]. PDMS is physiologically inactive, very low in toxicity, presents no health hazards, and is inexpensive [15]. Soft lithography is a simple, low cost and scalable way to fabricate micro- or nanostructures on a PDMS surface to make it become a superhydrophobic surface (WCA > 150°) similar to a lotus-leaf surface [16]. PDMS is a material which suitable for soft lithography due to its low shrinkage rates and easy to penetrate to micropattern materials [17]. However, PDMS does not have high enough strength properties and when PDMS was fabricated with micropatterns of soft lithography the resultant micropatterns would collapse [15,18]. The stiffness of PDMS depends on the degree of crosslinking agent; the higher degree of PDMS network’s crosslinking, the higher its stiffness [18,19,20]. Furthermore, the mechanical properties of PDMS can also be improved using nanofiller; the tensile modulus of the PDMS will increase when increasing the amount of silica in the PDMS [21]. However, nanosilica were prepared via the solgel technique, which was a complicated process and involved many chemicals in its synthesis, and which modified silica surfaces such as tetraethyl orthosilicate, ammonium hydroxide solution, ethanol, silane coupling agent and toluene [21]. 

Polymer blending is a simple, inexpensive technique which is commercially feasible for preparing products with unique properties [22]. The combination of two or more polymers will give the obtained materials unique properties and also provide an economical way to produce new materials [23]. In previous work, neat PDMS fabricated on micro-patterning from soft lithography was easy to collapse under external forces (Van der Waals force) [24]. To solve this problem, in this paper, polyurethane with non-toxic, non-flammable, environmentally friendly, economic manufacturing and good mechanical properties [25] was blended with PDMS to produce an antifouling film. Moreover, the performance of the PDMS:PU blend on antifouling performance and its ability for use in soft lithography to achieve higher hydrophobicity were studied [26].

## 2. Materials and Methods

### 2.1. Material and Chemicals

Polydimethylsiloxane elastomer (PDMS) used in this experiment had a density of 1.03, supplied by Dow Corning under the tradename of Sylgard 184. It had a two-part chemical (Part A as a base and Part B as a curing agent) containing the silicone base and a curing agent. Part A contained dimethyl siloxane, dimethylivinyl terminated, dimethylvinylated and trimethylated silica, tetra (trimethoxysiloxy) silane and ethyl benzene. The curing agent (Part B) contained dimethyl, methylhydrogen siloxane, dimethyl siloxane, dimethylvinyl terminated, dimethylvinylated and trimethylated silica, tetramethyl tetravinyl cyclotetra siloxane, and ethyl benzene. Polydimethylsiloxane elastomer (PDMS) substrates were blended by completely mixing 10 parts base to 1 part curing agent. Polyurethane (PU) with density 1.036 was procured from Smooth-On, USA. It had two-parts: Part A and Part B. Part A is 4,4′ methylenedicyclohexyl diisocyanate and part B is glycol and phenylmercury.

### 2.2. Preparation of the Polymer Blend

Blends of polydimethylsiloxane elastomer and polyurethane were prepared using an overhead stirrer (IKA Overhead Stirrers RW 20 digital). The ratios of PDMS:PU used in this study were 100:0, 95:5, 90:10, 10:90 5:95, and 0:100. A typical blending process was carried out by first preparing a polyurethane mixture with a mixing ratio of 10A:9B. The mixture of polyurethane was prepared using a mechanical stirrer at 250 rpm. Then, the mixture of polydimethylsiloxane elastomer was added to polyurethane and mixed at 250 rpm for 10 min. After that, the polymer blend was poured into a mold. During the process, air bubbles might occur, and a degassing step was required by placing the samples in a vacuum desiccator for 30 min to remove any small and large bubbles on the surface of the sample. Next, the sample was cured by heating in an oven at 60 °C for 3 h. Finally, after cooling the obtained samples were removed from the mold. 

### 2.3. Characterization of the Polymer Blend

#### 2.3.1. Scanning Electron Microscope (SEM)

To investigate phase morphology of the cross-section of the polymer blends’ films, a scanning electron microscope (FE-SEM, Hitachi S-4800) was employed. In this study, the polymer blend film samples were prepared by immersing them in liquid nitrogen for 3–5 min. Then, the sample was broken between two grips. Next, the specimens were coated with platinum under vacuum before observation by using accelerating voltage of 10 kV. The SEM/EDX studies were also performed for the material identification and phase dispersion. 

#### 2.3.2. Fourier Transform Infrared Spectra (ATR-FTIR)

FTIR analysis was performed with a Thermo Scientific/Nicolet Nexus IS5. FTIR spectra were collected between 4000 cm^−1^ to 650 cm^−1^ with a signal-to-noise ratio of 40,000:1 at spectral resolution of 0.45 cm^−1^. The IR spectra were used to characterize the functional groups and chemical vibrational frequencies of the polymers. 

#### 2.3.3. Contact Angle Measurement

Static contact angle measurements were performed using a Krȕss (model DSA 10) contact angle measuring instrument at ambient temperature to determine the wettability change of the polymer blend surfaces. A 10 μL sessile droplet of de-ionized water was vertically dropped, with a microsyringe, onto the surface of the samples. The images of the drop shape on the surface of the samples were captured using a camera equipped with a magnifying lens.

#### 2.3.4. Mechanical Test (LLOYD)

Mechanical properties such as tensile strength, modulus and elongation at break of the two polymers and the polymer blends were measured using universal testing machine with standard as per ASTM 882. The speed of stretching was 50 mm/min with dimensions of 100 mm × 10 mm × 1 mm. For each sample, an average result obtained from five individual specimens was reported. 

#### 2.3.5. Atomic Force Microscopic (AFM) 

Atomic force microscopy (AFM) significantly measures the local properties of materials and intermaterial interactions with nanoscale spatial and piconewton force resolution. The spatial sensitivity of the AFM photodetector was calibrated against a clean silicon wafer. Thin film sample of approximate 1 mm thickness was prepared from different ratio of PDMS:PU. For each sample, the topology and elastic modulus were measured from five different places over sizes of 1 × 1 × 0.1 cm with the scanAsyst-Air probe using PeakForce QNM modes with 0.4 N/m spring constant. The Peak Force Quantitative Nanomechanical Mapping (PFQNM) mode, also known as quantitative nanomechanical mapping, is a semicontact AFM mode, which constructed height and phase images simultaneously. Phase image acknowledged differences in modulus.

#### 2.3.6. Swelling Test

Swelling test was carried out to determine swelling properties of the samples when immersed in diverse mediums, following ASTM D3616. Swelling properties were calculated from variation in mass of the samples. The test specimens of weight, *M_o_*, (g) were immersed in DI water and seawater in bottles at room temperature for 24 to 240 h. The seawater used in the analysis was collected from the Gulf of Thailand at Koh Sichang, Chonburi province, Thailand, and it has the average salinity of ocean water, around 33–34 ppt with pH 7.7–8.0. After the required immersion time, the samples were removed, gently wiped and dried to eliminate any excess liquid, and the swollen samples were reweighted, M, (g). The degree of swelling was calculated using Equation (1).
(1)$$Q=\frac{M-M_{o}}{M_{o}}\times 100.\;$$
where: *Q* is degree of swelling

*M_o_* is the initial weight of sample (g)

*M* is the swollen weight of sample (g)

#### 2.3.7. Microfouling Analysis 

The barnacle adhesion study was carried out to determine level of microfouling by immersing the samples in a marine environment and determined by visual inspection as described by ASTM D3623. Barnacles were collected from seawater in the Gulf of Thailand at Koh Sichang Marine Science Research Centre of Chulalongkorn University (13°09′10.6″ N 100°49′02.6″ E), which has high barnacle population because breeding and broadcast-spawning frequently occur during this period, for 8 weeks during April to June [27]. The reproduction of barnacles increases because it is related to the warmer seawater temperatures. The reference specimen used in this study was SS400 carbon steel. SS400 steel sheets are typically used for general structures, and they are also used in marine ship structures, high-rise buildings, etc. [28]. For the field tests in a seawater environment, all samples were set in Teflon frames with 12-channel (see Figure 1). The size of the Teflon frame was 0.6 × 0.6 m^2^. Then, the frames were set on 2 different sides: facing the sea (uprush) and facing the shore (backwash) at a depth of 3 m from the seawater level (Figure 1) [24]. Subsequently, the number of barnacles on each sample surface were measured and captured by a digital still camera. 

Barnacles of average dimensions, i.e., between 5 to 20 mm in diameter and at least 30 mm in height, were selected for this method. The base area of each barnacle, A, (square meters) was approximated from an average base diameter, d_o_, (meters). The base area of each barnacle was calculated according to Equation (2).
(2)$$A=\frac{1}{4}\pi d_{o}^{2}.\;$$

The adhesion force measurement for all samples when barnacles settled on the surface was carried out on a digital force gauge (Inspex IPX-808) at room temperature. The capacities of the force measuring device are between 0 and 150 N (0 to 34 lb) to an accuracy of ±0.5 N (±0.1125 lb). Furthermore, the adhesive strength, τ, (pascal, Pa) of the two fouling release surfaces was evaluated according to ASTM D5618-94 by measuring the shear force, F, (newton, N) required to remove the barnacle by the base area, A, (square meters) of the barnacle as in Equation (3):(3)$$\tau =\frac{F}{A}$$

## 3. Results

### 3.1. Characteristics of PDMS:PU Blend Film 

#### 3.1.1. FTIR Spectral of PDMS:PU Blend Film

The chemical structures of the PDMS:PU blend film were verified using the FTIR spectra. To identify polydimethylsiloxane (PDMS), after polyurethane (PU) was added, attenuated total reflectance infrared spectroscopy was employed (Figure 2). The PDMS film displayed a characteristic Si-CH_3_ peak around 1250 cm^−1^, Si-O-Si at 1055 cm^−1^ (Si-O stretching), Si-CH_3_ at peak around 790 cm^−1^ (CH_3_ rocking and Si-C stretching) and asymmetric CH_3_ at 2962 cm^−1^. The FTIR spectrum of the polyurethane film showed NCO-termination at 2200 cm^−1^, -NH deformation at 1513.7 cm^−1^ and C-O-C ether group at 1160 cm^−1^: the C-H symmetric and asymmetric stretching vibrations of CH_2_ group were observed at 2923 cm^−1^ and 2850 cm^−1^, respectively. The PDMS:PU blend film with ratio 5:95 and 10:90 showed a strong peak of the N-H band at 3200–3400 cm^−1^ and a carbonyl peak (C=O) at 1700 cm^−1^, which represent PU’s characteristics in a polymer blend. On the other hand, the PDMS:PU blend film with ratio 90:10 and 95:5 showed a very weak peak of the N-H band at 3200–3400 cm^−1^ and of the carbonyl peak (C=O) at 1700 cm^−1^ due to the very small amount of PU in the blend. The obtained FTIR spectra support the proposed structure of the final PDMS:PU blend film.

#### 3.1.2. Water Contact Angle (WCA) of PDMS:PU Blend Film

To achieve antifouling characteristics, the surface characteristics of the polymer film, measured by water contact angle (WCA), should be greater than 90°, which is characteristic of hydrophobic surfaces. The contact angle from the water droplets on the PDMS film surface had a value of 109.5° ± 0.3° and PU film surface was 86.9° ± 3.2° (see Table 1). The PDMS film showed hydrophobic properties because the WCA value was higher than 90° [29]. When PMDS was blended with PU, it was found that the PDMS:PU blend film showed a higher value of WCA than the PU film. However, the water contact angle of the PDMS:PU blend film decreased from 103.4° ± 3.8° to 91.4° ± 0.8° when the PU content was increased from 5 wt.% to 95 wt.% of PU. It seemed that the PDMS:PU blend film became more hydrophilic than the PDMS film due to the hydrophilic group (glycol) of the PU backbone [30]. Based on the results, it was concluded that the ratio of the PDMS:PU blend film at 95:5 exhibited high hydrophobic surface properties since it presented a water contact angle of larger than 100°. 

#### 3.1.3. Morphology of the Polymer Blend Film

The morphology of the polymer blends between PDMS and PU were observed by using SEM. The neat PDMS film cross-section showed a rough surface (Figure 3a) which was caused by the ductile material fracture, but on the other hand, the morphology of polyurethane (Figure 3b) showed a smooth surface due to brittle material fracture. The brittle properties of a material can normally be identified by the smoothness of the fracture surface of the cross-section, such as observed in polyurethane, and the ductile properties can be observed by the roughness of the fracture surface cross-section, such as observed in polydimethylsiloxane [31]. The ratio of the polymer blends of PDMS:PU; 95:5 and 5:95 showed spherical particles of polyurethane, as observed in Figure 3c,d. The SEM image of PDMS:PU blend (95:5) film (Figure 3c) clearly illustrates good distribution of PU particles in PDMS matrix, with the average diameter of the dispersed PU particles at about 8.3 ± 5.6 µm. 

Energy-dispersive X-ray spectroscopy (EDX) was carried out to study the elemental distribution of PDMS, PU and the PDMS-PU blends (see Figure 4). It could be observed that the silicon (Si) weight percentage of the PDMS:PU blend decreased from 57.35 wt.% to 32.29 wt.% while more carbon atoms were detected with the addition of polyurethane into the system. It also illustrated that the carbon (C) weight percentage of the surface rose from 24.50 wt.% to 53.47 wt.% with the increase of polyurethane up to 95%. The results confirmed that polyurethane was distributed in PDMS matrix (Table 2) [32].

#### 3.1.4. Mechanical Properties Studies of the PDMS:PU Blend Film

The PDMS film showed an elastic property with a tensile strength and Young’s modulus of 1.14 ± 0.29 MPa and 1.50 ± 0.18 MPa, respectively. After blending PDMS with PU, the mechanical properties of the PDMS:PU blend films were increased due to the introduction of the more rigid polyurethane. As shown in Figure 5, for the PDMS:PU blend film when PU content increased from 0, 5, 95, and 100, Young’s modulus increased 1.50 ± 0.18 MPa (PDMS film), 1.75 ± 0.35 MPa, 863.78 ± 46.49 MPa, and 1559.84 ± 266.16 MPa, respectively, and the PDMS:PU (95:5) blend film achieved an acceptable mechanical properties and still retained good deformation resistance [33] which was suitable for micropattern fabricating using the soft lithography process. 

#### 3.1.5. Atomic Force Microscopic (AFM) Studies

Atomic force microscope (AFM) was employed to collect topography images and measure the roughness of the sample scanned. The AFM height sensor images displayed in Figure 6a,c,e represent the PDMS:PU blend (95:5), PDMS:PU blend (5:95) and polyurethane film, respectively. The polyurethane film showed a relatively smooth and uniform surface. On the other hand, the PDMS:PU blend film showed a rough surface. Thus, different topographical features (both nanoscale and macroscale) were observed on PDMS:PU blend films with different mixing ratios. 

The AFM DMT images displayed in Figure 6b,d,f represent the PDMS:PU blend (95:5), PDMS:PU blend (5:95) and polyurethane, respectively. The indentation modulus among the samples was calculated from the QNM^tm^ mode in PFQNM. In the AFM topography images of the film, the surface showed both a brighter phase and a darker phase, with the brighter phase showing a higher modulus than darker phase [34]. The AFM topography images of the PDMS:PU blend film presented the dark areas as polydimethylsiloxane and the bright areas as polyurethane. The relative modulus of the material surfaces of PDMS:PU (95:5) blend was 123 ± 13 mArb. However, for PDMS:PU blend (5:95) film, the modulus increased to 160 ± 34 mArb, which was a 30% higher modulus than PDMS:PU blend (95:5). The relative modulus of the PU film was 180 ± 30 mArb, and the modulus tended to increase with increase PU ratio [35]. Therefore, the blending of PDMS and PU could improve the surface modulus of the PDMS:PU blend.

#### 3.1.6. Effect of Swelling

The applicability of polymer films for antifouling products is based on the bond breakage of samples when immersed in different testing mediums, which can be predicted by swelling behavior and water resistance in a natural environment. In this paper, polydimethylsiloxane elastomer, polyurethane and the PDMS:PU blend at 95:5 blend ratio were studied for their swelling behaviors. The mechanism of swelling is represented by two properties; the addition of weight and the reduction of sample weight. The final sample would shrink or damage for reduction conditions [36]. The penetration of DI water and seawater (SW) through the polymer was observed. After immersion for 72 h, each type of sample began to lose weight, illustrating hydrolysis degradation [37]. However, seawater immersion is a complicated medium due to the microorganisms and elements contained in sea water. The PDMS film immersed in seawater showed an increase in the rate of mass loss by enzymatic biodegradation [38] when compared with tests carried out in DI water. This was because the seawater contained microorganisms such as bacteria, fungi, algae and plankton. The highest degree of swelling was 2.12 in 72 h and over 5 days the mass loss was slightly reduced. On the other hand, the mass loss of PU in the two mediums showed that the degree of swelling of PU slightly increased. Then, the steadiness degree of swelling was accomplished after approximately 72 h in DI water and sea water. Mass loss for the PDMS:PU blend (95:5) increased with an increase in immersion time and then it remained steady. However, the polymer blend of PDMS:PU blend (95:5) exhibits a degree of swelling close to PDMS at 2.36 (Figure 7).

#### 3.1.7. Barnacle Measurements

The short-term antifouling measurement of the polymer film was evaluated in seawater at Koh Sichang Marine Science Research Centre of Chulalongkorn University (13°09′10.6′′ N 100°49′02.6′′ E) during April to June. The antifouling behaviors of polydimethylsiloxane, polyurethane and the PDMS:PU blend (95:5) were compared with carbon steel (CS) as the reference material, with an area of 6 cm × 6 cm, using a digital microscope as shown in Figure 8. The reference carbon steel started to corrode and rusting was observed on the surface after being immersed in seawater for 2 weeks. The growth rate of the barnacles on the surface of the carbon steel was much higher than the other studied surfaces. To improve the antifouling properties of the carbon steel, a blend of PU and PDMS was studied as a polymer film. From the study, it was clearly observed that surface of the PDMS film and PDM:PU blend (95:5) film clearly improved the resistance to fouling of the surface (Figure 9). 

After 2 weeks, it was clearly observed that barnacles on the surface of the carbon steel had rapidly increased and partially covered all surfaces at approximately 2.5 ± 0.4 marine barnacles/cm^2^ (see Figure 10). The marine organisms on the carbon steel samples, which were placed in the sea facing the shore and facing away from the shore, decreased from 2.5 ± 0.4 to 0.7 ± 0.2 marine barnacles/cm^2^ and 2.7 ± 0.8 to 1.0 ± 0.2 marine barnacles/cm^2^ after 4 and 8 weeks, respectively, which was due to the increase in the size of the barnacles. After 2 weeks of immersion of the PDMS film in sea water, the number of barnacles was rather low initially; biofouling on the surface of samples facing the shore side was approximately 0.5 ± 0.1 marine barnacles/cm^2^, while that for the samples facing away from the shore was approximately 0.9 ± 0.1 marine barnacles/cm^2^. When immersed longer, some biofouling disappeared from the surface of the PDMS film since the fouling was released more easily from the surface of the silicone samples due to weak adhesion between the surfaces [39]. Due to its more hydrophilic properties, the surface of the PU film had more barnacles attached than the PDMS, but it had slightly fewer barnacles attached than the carbon steel. The marine organisms collected after 2 weeks were 1.2 ± 0.7 marine barnacles/cm^2^. The PDMS:PU blend (95:5) samples exhibited a lower number of barnacles than the carbon steel and the PU film. After 2 weeks of immersion, the number of marine barnacles on the PDMS:PU blend (95:5) film facing the shore side was 0.5 ± 0.2 marine barnacles/cm^2^ and the PDMS:PU blend (95:5) film facing away from the shore was 1.1 ± 0.3 marine barnacles/cm^2^, which were close to the PDMS surfaces. For 4–8 weeks, the polymer blend samples showed the best result for the number of barnacles on the surface, and the different of antifouling characteristics between the PDMS:PU blend (95:5) samples and the carbon steel were clearly noticeable. 

Moreover, since the study focused on antifouling surfaces, which is how easily adhered marine organisms were removed from the surface without toxicity to marine organisms, barnacle adhesion strength measurements were also calculated by dividing the adhesive force by the barnacle base area. The results indicated that PDMS and the PDMS:PU blend (95:5) film showed good antifouling properties because of the low attachment number and low adhesive strength of the barnacles on the surface of these materials. This therefore means that the surface of the PDMS and the PDMS:PU blend (95:5) film have excellent antifouling properties. Therefore, the PDMS film and the PDMS:PU polymer blend exhibited lower adhesive strength than carbon steel, by around 80–90%, after immersion in seawater for 4 and 8 weeks (Figure 11). Therefore, from the study we could conclude that barnacles favored attachment to hydrophilic surfaces, i.e., carbon steel and PU film. Our results correspond to the data from Finlay et al. [40] which suggested that hydrophilic surfaces or higher energy surfaces increased barnacle attachment. 

### 3.2. Characteristics of PDMS:PU Blend Film with Micro Patterning Fabricated by Soft Lithography

#### 3.2.1. Morphology of PDMS:PU Blend Film with Micro Pattering Fabricated by Soft Lithography

Microstructures or micro patterning were fabricated on the surface of the PDMS:PU blend (95:5) by the soft lithography process to further increase the hydrophobic properties of the polymer blend film. The microstructures or micropatterns on the PDMS:PU blend (95:5) surface were observed by using SEM (Figure 12). The side view using of the PDMS:PU blend (95:5) film (see Figure 12a) clearly illustrated uniform sharklet structures on the surface, suggesting that the PDMS:PU blend (95:5) was able to be further modified by the soft lithography process. 

#### 3.2.2. Water Contact Angle (WCA) of Polymer Blend with Micro Patterning Fabricated by Soft Lithography 

The hydrophobicity of the PDMS:PU blend (95:5) film with micro patterning by the soft lithography process was determined using the water contact angle measurement. The contact angle from the water droplets on the PDMS:PU blend (95:5) surface without micro-structures has a value of 103.4° ± 3.8° and the water contact angle on PDMS:PU blend (95:5) film with micro patterns on the surface showed improved hydrophobicity with a water contact angle (WCA) of 128.8° ± 1.6°. The result indicated that the hydrophobicity of the PDMS:PU blend (95:5) film could be further improved by the soft lithography process (Table 3).

## 4. Conclusions

A nontoxic material with simple preparation process and economic viability with antifouling film performance was prepared from a polymer blend with different ratios of PDMS and PU. Based on the hydrophobicity of the film, the PDMS:PU blend (95:5) showed the best results at 103.4° ± 3.8°. When PU was introduced to PDMS, the phase and morphology of the polymer blend clearly illustrated a good distribution of the PU in PDMS. In addition, the mechanical properties resulting from stress–strain curve and PF-QNM revealed the stiffness differences among the samples with different mixing ratios since polyurethane has been recognized to increase hardness and modulus when blended with PDMS. From the results, the antifouling performance of PDMS:PU blend (95:5) showed a low number of barnacles on the surface and a low adhesive strength of 0.07 MPa for 8 weeks, or a 80–90% decrease when compared to the reference carbon steel sample, and the biofouling could easily be removed from the surface. The result showed that the PDMS:PU blend (95:5) exhibited excellent antifouling properties, making it suitable to be applied as a polymer coating in the marine environment. In this study, it was also demonstrated that soft lithography process could be employed to further improve the hydrophobicity of the film. The PDMS:PU blend (95:5) showed small spherical droplets of polyurethane which could penetrate into a 3 µm width sharklet pattern on the soft lithography mold and further improve the hydrophobicity of the polymer blend film, with an improved water contact angle (WCA) of 128.8° ± 1.6°. 

## Figures and Tables

**Figure 1 polymers-13-02242-f001:**
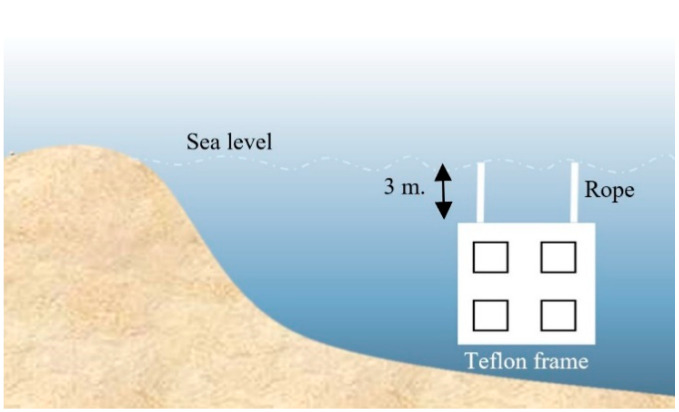
Experiment setup of microfouling analysis.

**Figure 2 polymers-13-02242-f002:**
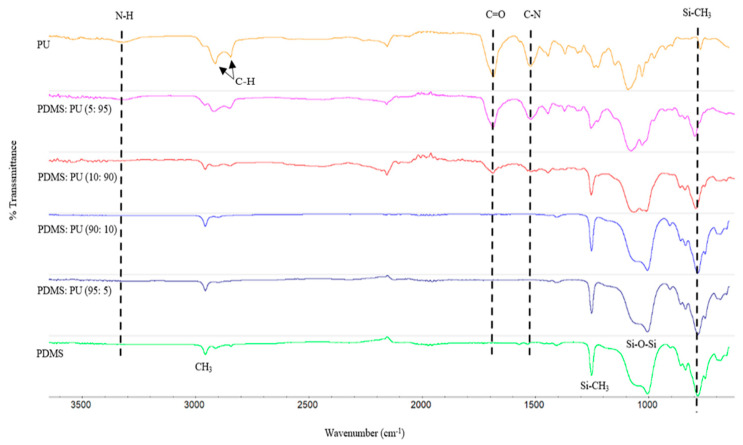
FTIR spectra of PDMS:PU blend films between 3500–500 cm^−1^.

**Figure 3 polymers-13-02242-f003:**
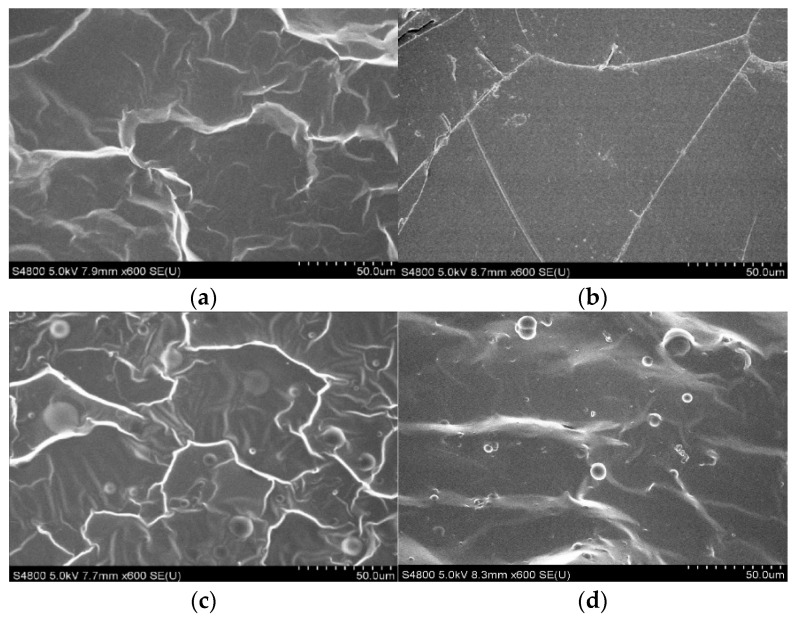
SEM image of cross-section view of (**a**) polydimethylsiloxane; (**b**) polyurethane; (**c**) PDMS:PU blend (95:5); and (**d**) PDMS:PU blend (5:95).

**Figure 4 polymers-13-02242-f004:**
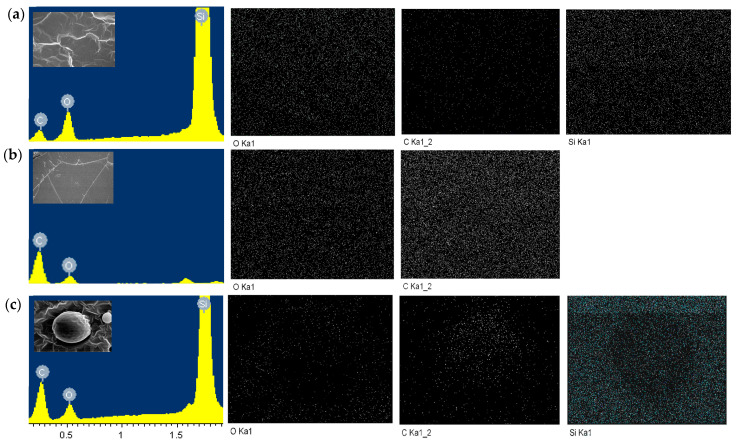
SEM-EDX image of: (**a**) polydimethylsiloxane; (**b**) polyurethane; (**c**) PDMS:PU blend (95:5). The graphs represent the relative atomic fractions of the three major elements.

**Figure 5 polymers-13-02242-f005:**
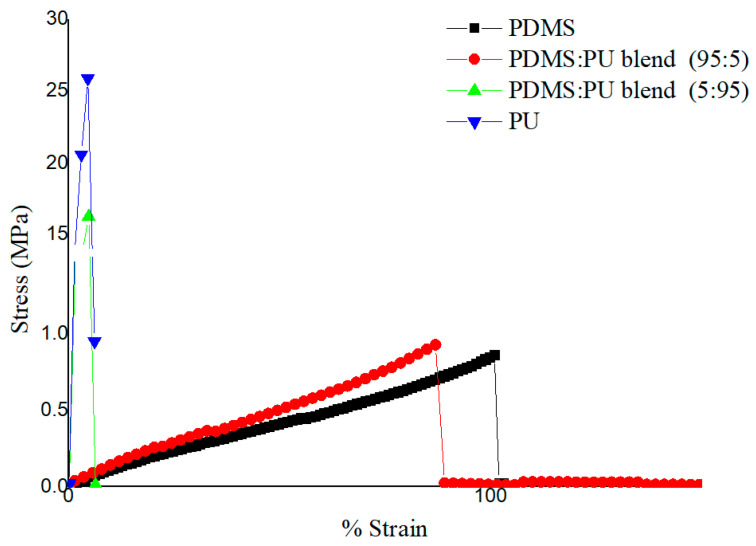
Stress–strain curves of polydimethylsiloxane, polyurethane, PDMS:PU blend (at room temperature).

**Figure 6 polymers-13-02242-f006:**
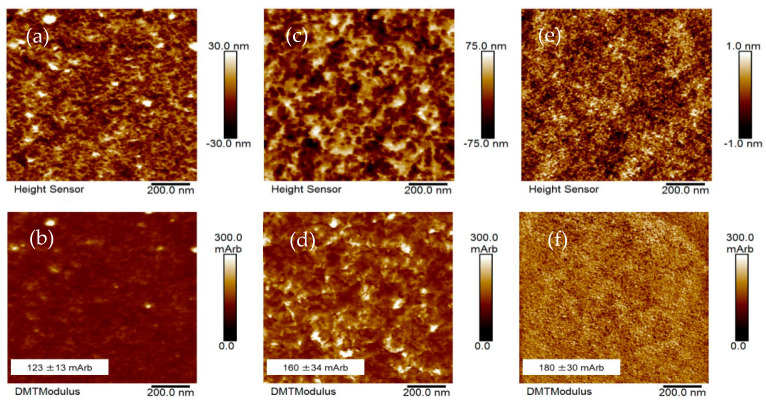
AFM topology images (**a**,**b**) PDMS:PU (95:5) blend; (**c**,**d**) PDMS:PU (5:95) blend; (**e**,**f**) polyurethane.

**Figure 7 polymers-13-02242-f007:**
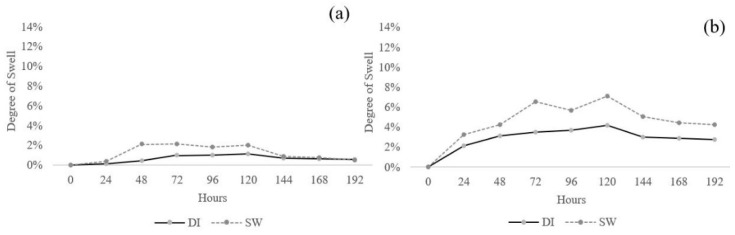
Time dependent mass loss in DI water and seawater of (**a**) polydimethylsiloxane, (**b**) polyurethane and (**c**) PDMS:PU blend (95:5).

**Figure 8 polymers-13-02242-f008:**
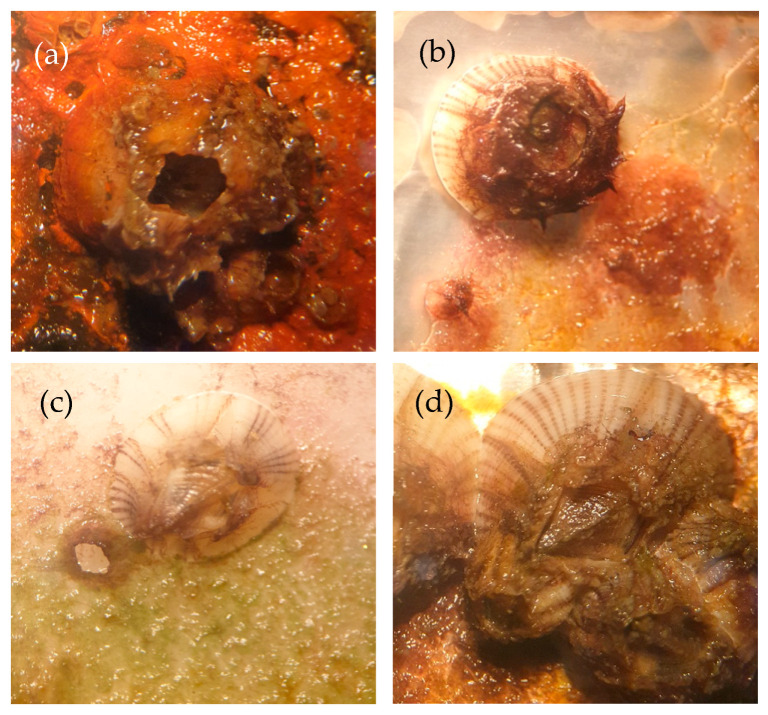
Digital microscope images of marine biofouling on various material surface in seawater (**a**) carbon steel, (**b**) polydimethylsiloxane, (**c**) PDMS:PU blend (95:5) and (**d**) polyurethane.

**Figure 9 polymers-13-02242-f009:**
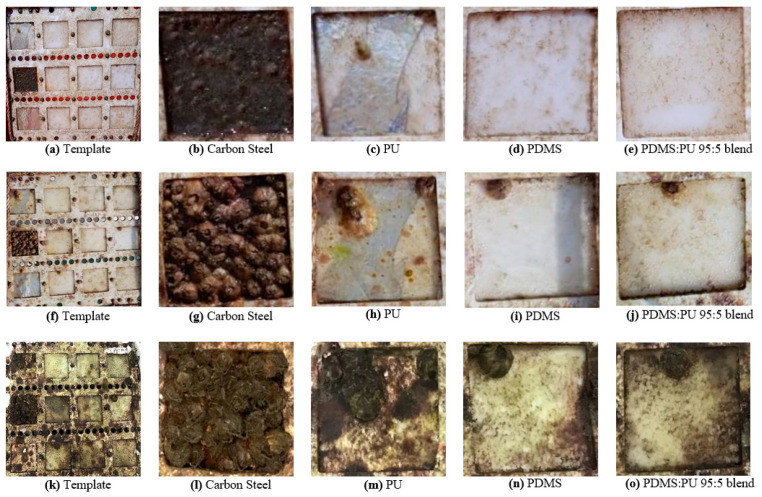
Marine biofouling on various sample surfaces after immersed in seawater environment (**a**–**e**) immersed for 2 weeks, (**f**–**j**) immersed for 4 weeks and (**k**–**o**) immersed for 8 weeks (in April-June).

**Figure 10 polymers-13-02242-f010:**
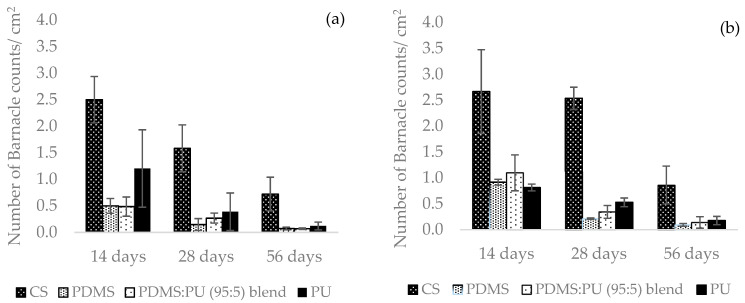
Barnacle counts after 2 weeks, 4 weeks and 8 weeks in seawater with different immersion duration: (**a**) facing the shore and (**b**) facing away from the shore.

**Figure 11 polymers-13-02242-f011:**
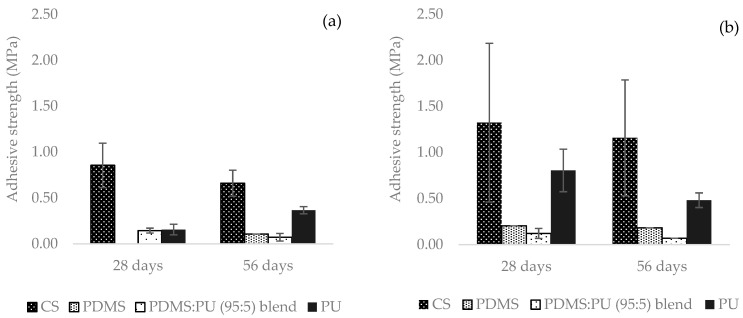
Adhesive strength (MPa) of barnacle on surface after 2 weeks, 4 weeks and 8 weeks in seawater: (**a**) facing the shore and (**b**) facing away from the shore.

**Figure 12 polymers-13-02242-f012:**
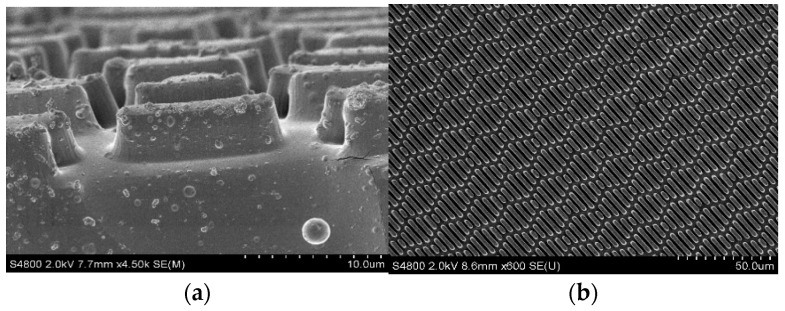
SEM image of (**a**) side view of the PDMS:PU blend (95:5) ridge sharklet pattern and (**b**) top-view SEM image of PDMS:PU blend (95:5) ridge sharklet pattern fabricated by soft lithography process.

**Table 1 polymers-13-02242-t001:** Water contact angle on various material surfaces polydimethylsiloxane; polyurethane; PDMS:PU blend (95:5, 90:10, 10:90, 5:95).

Polyurethane	PDMS:PU blend (5:95)	PDMS:PU blend (10:90)
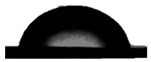 86.9° ± 3.2°	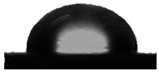 91.4° ± 0.8°	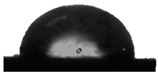 92.2° ± 2.5°
PDMS:PU blend (90:10)	PDMS:PU blend (95:5)	Polydimethylsiloxane
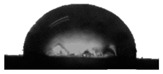 94.6° ± 0.8°	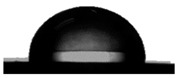 103.4° ± 3.8°	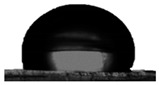 109.5° ± 4.2°

**Table 2 polymers-13-02242-t002:** EDX results (weight percentage) obtained on each type of materials.

Element	Polydimethylsiloxane		PDMS:PU Blend (95:5)			Polyurethane		
	Weight%	Atomic		Weight%	Atomic		Weight%	Atomic
C	24.30	38.82		53.47	68.58		66.05	72.16
O	18.35	22.00		14.24	13.72		33.95	27.84
Si	57.35	39.18		32.29	17.71		0	0

**Table 3 polymers-13-02242-t003:** Water contact angle on various material surfaces PDMS:PU blend (95:5) and PDMS:PU blend (95:5) with sharklet pattern.

PDMS:PU blend (95:5)	PDMS:PU blend (95:5) sharklet pattern
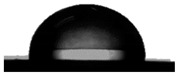 103.4° ± 3.8°	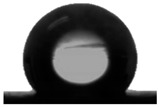 128.8° ± 1.6°

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
