# Peer review of "Simple Preparation of Polydimethylsiloxane and Polyurethane Blend Film for Marine Antibiofouling Application"

_polymers, 2021, doi:10.3390/polym13142242_

Round 1
Reviewer 1 Report
In this study, the authors claimed that a superhydrophobic polymeric film for anti fouling application made from a blend of different eco-17 friendly materials was investigated. However, according to the results, the water contact of film low 150°ï¼Œ that is, hydrophobicity (the max is smaller that 130°, thus it is not superhydrophobicity). In addition, the PDMS:PU blend ratio was mussed, such as in the Figure 2, there are two ratio, while in the Figure 4, there are four groups; in the table 1 only one ratio. Therefore, the authors should better to organize the paper.
Author Response
Dear Reviewer, here is the revised manuscript. We corrected it according to the recommendations of the reviewers, as is specified in the attached document. We hope that the manuscript is now satisfactory and suitable for publication.
Point 1: In this study, the authors claimed that a superhydrophobic polymeric film for antifouling application made from a blend of different eco-17 friendly materials was investigated. However, according to the results, the water contact of film low 150°, that is, hydrophobicity (the max is smaller that 130°, thus it is not superhydrophobicity).
Response 1: We have revised it (line 15-19) as followed:
“A key to prevent undesirable fouling of any structure in the marine environment, without harming any microorganisms, is to use a polymer film with high hydrophobicity. The polymer film which simply prepared from a blend of hydrophobic polydimethylsiloxane elastomer and hydrophilic polyurethane showed improved properties and economic viability for antifouling film for the marine industry.”
Point 2: In addition, the PDMS:PU blend ratio was mussed, such as in the Figure 2, there are two ratio, while in the Figure 4, there are four groups; in the table 1 only one ratio. Therefore, the authors should better to organize the paper.
Response 2: Thank you for your suggestion. We have already revised them as recommended in the manuscript.

Reviewer 2 Report
Why such strange ratios of PDMS to PU were used? Why 95:5 and 5:95 and not 50:50 or 25:75 and 75:25? Why actually 5%? Any previous results, references?
Authors are writing that Figure 2 shows cross-section of materials. How was this cross-section prepared?
It is nothing new that when ratio of 2 components is 95:5, then minor phase is disperesed in matrix. Please include more conclusions in the SEM description.
Please check again the description of the FTIR spectra of polyurethane and enhance it with the more bands showing urethane groups.
Why Figure 4 shows spectra for samples, which were not mentioned earlier in the experimental section?
Same for Table 2.
Considering the description of WCA results, please refer to the structure of polyurethane.
What means 455 and 464 in Figure 8?
Generally the quality of the paper is poor in terms of its organization.
Also in discussion please refer more to the other works.
Author Response
Dear Reviewer, here is the revised manuscript. We corrected it according to the recommendations of the reviewers, as is specified in the attached document. We hope that the manuscript is now satisfactory and suitable for publication.
Point 1: Why such strange ratios of PDMS to PU were used? Why 95:5 and 5:95 and not 50:50 or 25:75 and 75:25? Why actually 5%? Any previous results, references?
Response 1:
- In our preliminary study, we looked at PDMS:PU blend ratios from 100:0, 95:5, 90:10, 80:20, 50:50, 20:80, 5:95 and 0:100 and the results showed that when polyurethane ratios were between 20-80 wt%, it was unable to form polymer blend. Hench we are only reporting the ratios that we were able to form the polymer blend.
- From our study, the ratios of PDMS:PU blend 95:5 and 5:95 were used to study the effect of hydrophobic against hydrophilic surfaces on antifouling characteristics.
- Moreover, we would like to retain the hydrophobicity of PDMS and improve the modulus strength of the polymer blend to prevent microstructures fabricated by soft lithography from collapsing by adding only a small amount of polyurethane into PDMS:PU blend.
Point 2: Authors are writing that Figure 2 shows cross-section of materials. How was this cross-section prepared?
Response 2: We have revised it (line 126-128) as followed:
In this study, the polymer blend film samples were immersed in liquid nitrogen for 3-5 minutes. Then, the sample was broken between two grips and the broken samples were placed upright on the stub.
Point 3: It is nothing new that when ratio of 2 components is 95:5, then minor phase is disperesed in matrix. Please include more conclusions in the SEM description.
Response 3: We have revised them as recommended
Point 4: Please check again the description of the FTIR spectra of polyurethane and enhance it with the more bands showing urethane groups.
Response 4: We have revised them as recommended.
Point 5: Why Figure 4 shows spectra for samples, which were not mentioned earlier in the experimental section? Same for Table 2.
Response 5: We have revised them as recommended.
Point 6: Considering the description of WCA results, please refer to the structure of polyurethane.
Response 6: We have revised them as recommended.
Point 7: What means 455 and 464 in Figure 8?
Response 7:
455 and 464 in Figure 8 refer to the line number of the manuscript. It is not part of the Figure.
Point 8: Generally the quality of the paper is poor in terms of its organization. Also in discussion please refer more to the other works
Response 8: Thank you for your kind suggestions. We have reorganized and revised it as recommended.

Reviewer 3 Report
Dear Author,
It is my great pleasure to review your manuscript about PDMS/PUR blend for Marine Antibiofouling application. Please find my comments below. Thanks!
- Contents does not align with the paper title. Consider to revise the title.
- Pay attention to the citations. A Bracket [ ] but not a parenthesis ( ) should be used. For example, ref. 1 should be [1], but not (1).
- Line 16-17. Rephrase the sentence.
- Line 18-20. Rewrite the sentence. The polymer film, which simply prepared from a blend of hydrophobic polydimethylsiloxane elastomer and hydrophilic polyurethane, showed improved properties…..
- Line 28, not clear. It was meant to compare the bonding strength of barnacles to the polymer film vs. to the carbon steel, right? rewrite that part of the sentence.
- Line 33-38. Add references.
- Line 33-46. Rewrite the sentences to make it easy to understand. Check the attached PDF file for details.
- line 55-57; 58-62 Add references.
- Line 71. lower ratio means less PDMS and more crosslinking agent, thus higher degree of crosslinking. then should not the material become more stiffer?
- Line 77-8. summary of the reference 14 is not clear. and what kind of process was used there? based on the info here, it does not sounds like complicated and involve lots of chemical.
- line 80-85. the content is repeated, rewrite these three sentences.
- Line 89-91. might be better to talk about a bit more about the methods and chemcials investigated, and bridge to the study of this manuscript.
- Line 105. Should “gravity” be DENSITY?
- Line 330-332. based on the structure of the PDMS and PUR used in the blending, their compatibility should be very poor. but FTIR could not provide the proof for the phase separation. for this purpose, DSC is
- Figure 4. The peaks in the range of 3000-2800 cm-1 for PUR spectra should be C-H stretching but not N-H stretching.
- Looks like the soft lithography process is the key to achieve the super-hydrophobic surface, but it was not clearly stated in the introduction part. Also, the reason why PUR was chosen is not clear.
- Line 352-54. how about the blends with less than 5% PUR? they all will have a water contact angle greater than 100. Based on your purpose, the study should be focused on the blends has a PUR content at least less than 50% instead of 5-95%.
- Table 2 and Figure 5 were both presenting the contact angle. This is not necessary.
- Reference section. correct the format of the references. use BOLD for all the Years. journal name should be abbreviation. and a comma should be used between year, volume, and pages. Check published paper from the journal home page for examples.
- Check other suggestion on the attached PDF file.

Author Response
Dear Reviewer, here is the revised manuscript. We corrected it according to the recommendations of the reviewers, as is specified in the attached document. We hope that the manuscript is now satisfactory and suitable for publication.
Point 1: Contents does not align with the paper title. Consider to revise the title.
Response 1: We revised the title as follows:
“Simple Preparation of Polydimethylsiloxane and Polyurethane Blend Film for Marine Antibiofouling Application”
Point 2: Pay attention to the citations. A Bracket [ ] but not a parenthesis ( ) should be used. For example, ref. 1 should be [1], but not (1).
Response 2: We have revised them as suggested.
Point 3: Line 16-17. Rephrase the sentence.
Response 3: We have revised them as suggested.
Point 4: Line 18-20. Rewrite the sentence. The polymer film, which simply prepared from a blend of hydrophobic polydimethylsiloxane elastomer and hydrophilic polyurethane, showed improved properties….
Response 4: Thank you for your suggestion. We have revised them (line 16-19) as suggested:
“The polymer film which simply prepared from a blend of hydrophobic polydimethylsiloxane elastomer and hydrophilic polyurethane showed improved properties and economic viability for antifouling film for the marine industry.”
Point 5: Line 28, not clear. It was meant to compare the bonding strength of barnacles to the polymer film vs. to the carbon steel, right? rewrite that part of the sentence.
Response 5: Thank you for your suggestion. We have revised it (line 26-28) as suggested.
“Result from a field test in the Gulf of Thailand illustrated that the bonding strength between barnacles and PDMS:PU (95:5) blend film (0.07 MPa) were lower than the bonding strength between barnacles and carbon steel (1.16 MPa). The barnacles on the PDMS:PU (95:5) blend film were more easily removed from the surface.”
Point 6: Line 33-38. Add references.
Response 6: We have added the references in manuscript as suggested.
Point 7: Line 33-46. Rewrite the sentences to make it easy to understand. Check the attached PDF file for details.
Response 7: We have revised them as suggested.
“Surfaces of a building or a boat in marine environment tend to accumulate high concentration of fouling attachments of a micro-macro organism biofilm. This is the major damage to the structures and the equipment.
Point 8: line 55-57; 58-62 Add references.
Response 8: We have added the references in manuscript as suggested.
Point 9: Line 71. lower ratio means less PDMS and more crosslinking agent, thus higher degree of crosslinking. then should not the material become more stiffer?
Response 9: We have revised it (line 74-76) as followed:
“The stiffness of PDMS depends on the degree of crosslinking agent, the higher degree of PDMS network’s crosslinking, the higher its stiffness [18] [19] [20].”
Point 10: Line 77-8. summary of the reference 14 is not clear. and what kind of process was used there? based on the info here, it does not sounds like complicated and involve lots of chemical.
Response 10: We have revised it (line 78-81) as follow:
“However, nano silicas were prepared via solgel technique which was a complicated process and involved many chemicals in synthesis process and modified silica surface, such as tetraethyl orthosilicate, ammonium hydroxide solution, ethanol, silane coupling agent and toluene [21].”
Point 11: line 80-85. the content is repeated, rewrite these three sentences.
Response 11: We have revised them as recommended.
Point 12: Line 89-91. might be better to talk about a bit more about the methods and chemcials investigated, and bridge to the study of this manuscript.
Response 12: We have revised them as recommended.
Point 13: Line 105. Should “gravity” be DENSITY?
Response 13: We have revised it (line 96) as followed;
“Polydimethylsiloxane elastomer (PDMS) used in this experiment had a density of 1.03, supplied by Dow Corning under the tradename of Sylgard 184.”
Point 14: Line 330-332. based on the structure of the PDMS and PUR used in the blending, their compatibility should be very poor. but FTIR could not provide the proof for the phase separation. for this purpose, DSC is
Response 14: Thank you for your suggestion. We have revised them as recommended.
Point 15: Figure 4. The peaks in the range of 3000-2800 cm for PUR spectra should be C-H stretching but not N-H stretching.
Response 15: We have revised the Figure 2 as suggested.
Point 16: Looks like the soft lithography process is the key to achieve the super-hydrophobic surface, but it was not clearly stated in the introduction part. Also, the reason why PUR was chosen is not clear.
Response 16: We have revised it (line 85-87) as follow;
The reason to add polyurethane (PU) into PDMS is because in previous work, when polymer film prepared from neat PDMS undergoes micro-patterning from soft lithography process, the micro pattern is easy to collapse under external forces (Van der Waals force) so we want to improve the mechanical strength of the polymer blend and retain the good hydrophobic properties of PDMS by blending with polymer of good mechanical property such as PU.
Point 17: Line 352-54. how about the blends with less than 5% PUR? they all will have a water contact angle greater than 100. Based on your purpose, the study should be focused on the blends has a PUR content at least less than 50% instead of 5-95%.
Response 17:
- In our preliminary study, we looked at PDMS:PU blend ratios from 100:0, 95:5, 90:10, 80:20, 50:50, 20:80, 5:95 and 0:100 and the results showed that when polyurethane ratios were between 20-80 wt%, it was unable to form polymer blend. Hench we are only reporting the ratios that we were able to form the polymer blend.
- From our study, the ratios of PDMS:PU blend 95:5 and 5:95 were used to study the effect of hydrophobic against hydrophilic surfaces on antifouling characteristics.
- Moreover, we would like to retain the hydrophobicity of PDMS and improve the modulus strength of the polymer blend to prevent microstructures fabricated by soft lithography from collapsing by adding only a small amount of polyurethane into PDMS:PU blend.
Point 18: Table 2 and Figure 5 were both presenting the contact angle. This is not necessary.
Response 18 Figure 5 was removed from the manuscript as suggested.
Point 19: Reference section. correct the format of the references. use BOLD for all the Years. journal name should be abbreviation. and a comma should be used between year, volume, and pages. Check published paper from the journal home page for examples.
Response 19: We have revised them as suggested.
Point 20: Check other suggestion on the attached PDF file.
Response 20: We revised them as suggested.

Round 2
Reviewer 1 Report
no comments
Author Response
Dear Editor, here is the revised manuscript. We have corrected it according to the previous recommendations of the reviewers, as is specified in the attached document. We hope that the manuscript is now satisfactory and suitable for publication.

Reviewer 2 Report
Everything in order after corrections.
Author Response
Dear Reviewer, here is the revised manuscript. We have corrected it according to the previous recommendations of the reviewers, as is specified in the attached document. We hope that the manuscript is now satisfactory and suitable for publication.
